# Characterization of JAK1 Pseudokinase Domain in Cytokine Signaling

**DOI:** 10.3390/cancers12010078

**Published:** 2019-12-27

**Authors:** Juuli Raivola, Teemu Haikarainen, Olli Silvennoinen

**Affiliations:** 1Faculty of Medicine and Life Sciences, Tampere University, 33014 Tampere, Finland; juuli.raivola@tuni.fi (J.R.); teemu.haikarainen@tuni.fi (T.H.); 2Institute of Biotechnology, Helsinki Institute of Life Science HiLIFE, University of Helsinki, 00014 Helsinki, Finland; 3Fimlab Laboratories, Fimlab, 33520 Tampere, Finland

**Keywords:** JAK, STAT, cytokine, cytokine receptor, cancer, inflammation

## Abstract

The Janus kinase-signal transducer and activator of transcription protein (JAK-STAT) pathway mediates essential biological functions from immune responses to haematopoiesis. Deregulated JAK-STAT signaling causes myeloproliferative neoplasms, leukaemia, and lymphomas, as well as autoimmune diseases. Thereby JAKs have gained significant relevance as therapeutic targets. However, there is still a clinical need for better JAK inhibitors and novel strategies targeting regions outside the conserved kinase domain have gained interest. In-depth knowledge about the molecular details of JAK activation is required. For example, whether the function and regulation between receptors is conserved remains an open question. We used JAK-deficient cell-lines and structure-based mutagenesis to study the function of JAK1 and its pseudokinase domain (JH2) in cytokine signaling pathways that employ JAK1 with different JAK heterodimerization partner. In interleukin-2 (IL-2)-induced STAT5 activation JAK1 was dominant over JAK3 but in interferon-γ (IFNγ) and interferon-α (IFNα) signaling both JAK1 and heteromeric partner JAK2 or TYK2 were both indispensable for STAT1 activation. Moreover, IL-2 signaling was strictly dependent on both JAK1 JH1 and JH2 but in IFNγ signaling JAK1 JH2 rather than kinase activity was required for STAT1 activation. To investigate the regulatory function, we focused on two allosteric regions in JAK1 JH2, the ATP-binding pocket and the αC-helix. Mutating L633 at the αC reduced basal and cytokine induced activation of STAT in both JAK1 wild-type (WT) and constitutively activated mutant backgrounds. Moreover, biochemical characterization and comparison of JH2s let us depict differences in the JH2 ATP-binding and strengthen the hypothesis that de-stabilization of the domain disturbs the regulatory JH1-JH2 interaction. Collectively, our results bring mechanistic understanding about the function of JAK1 in different receptor complexes that likely have relevance for the design of specific JAK modulators.

## 1. Introduction

Janus kinases (JAK1–3 and Tyrosine Kinase 2, TYK2) are non-receptor tyrosine kinases that play a critical role in cell signaling via type I/II cytokines and interferons (IFNs) [1]. JAKs are constitutively bound to the intracellular part of the receptors that dimerize after ligand binding, thus enabling the transphophorylation and activation of JAKs and subsequent phosphorylation and translocation of STATs (signal transducer and activator of transcription proteins) into nucleus to regulate transcription. Structurally all JAKs consist of four domains; the N-terminal FERM (4.1-band, ezrin, radexin, moiesin) domain that together with the SH2-like domain (SH2) compose the main receptor binding moiety, followed by the C-terminal JAK-homologue (JH) 2 and JH1 domains. JH1 is an active tyrosine kinase while JH2 has an important role in regulating both basal and cytokine-induced activation [2,3,4,5]. Some residues required for the phosphotransferase reaction from ATP to a substrate are not conserved in JH2, such as altered Gly-rich region and lack of catalytic Asp, hence giving the domain “pseudokinase domain” designation [6]. Despite the absent or low catalytic activity, JH2s regulates JAK activity through allosteric mechanisms. For example, JH2 forms an autoinhibitory interaction with JH1 by binding to the hinge region and stabilizing the inactive conformation of JH1 [7]. Furthermore, the ATP binding to JH2 stabilizes the domain and nucleotide binding is critical for pathogenic activation of JAK2 [8,9]. In TYK2, a small ATP pocket-binding compound has been shown to efficiently decrease the activation [10]. In addition, the SH2-JH2 linker is an important regulative region that controls the activity of JAKs [4]. 

JAK1, as well as other JAK family members, harbor pathogenic gain-of-function mutations (GOFs) that cause acute lymphoid and -myeloid leukemia (ALL and AML, respectively). JAK1 mutations are found in varying types of cancers; e.g., 9% of hepatocellular carcinoma patients have been found to have JAK1 mutations [10]. In addition to GOFs, JAK loss-of-function mutations (LOFs) have been identified in JAK3, which cause severe combined immune deficiency (SCID); a disease resulting a depletion of B-cells and complete loss of T- and NK-cells [11,12]. A majority of the pathogenic mutants clusters in JH2 highlighting the regulative role of the domain [10]. The most common mutation, JAK2 V617F also resides in JH2. JAK2 V617F and accounts for ~95% of patients with polycythaemia vera and about 50% of patients with essential thrombocytosis and primary myelofibrosis [13,14,15]. Mechanistically, the mutation stabilizes JAKs the αC-helix (αC) in the N-lobe of JH2, and induces cytokine independent dimerization of the receptors, possibly via JH2-dimerization [16,17]. The αC resides in the JH1-JH2 interface but also lines the ATP-binding pocket. The region is important in cytokine-induced activation of kinases, and modulating it with mutations can inhibit constitutive activation of JAKs [17,18,19]. However, the mechanisms of function for many mutations is yet unknown.

Homologous mutation to JAK2 V617F in JAK1 (V658F) causes acute lymphoblastic leukemia [20]. In addition to the JAK2 V617F and its homologues, a distinctive cluster of GOFs resides in the JH1-JH2 interface and these mutations disrupt the autoinhibitory interaction between the domains. For example, JAK1 R724E has one of the highest incidence rates among the JAK1 pathologic mutations. It resides in the N-lobe of JH2 (similarly to homologous JAK2 R683S) and interacts with JH1. Other JAK1 mutants in the JH1-JH2 interface are A634D and L653F [21,22]. 

Currently there are three JAK JH1-targeted inhibitors in clinical use, but recently targeting JAK JH2s with small molecular compounds has gained interest as a potential treatment for constitutively active cytokine signaling [10,23,24]. The unique mode of ATP binding in JH2 may allow increased specificity over other eukaryotic protein kinases [25]. 

Although the general principles of the JAK-STAT signaling are well established, the underlying differences between the function of JAKs and their individual domains in different cytokine signaling pathways are not fully defined. Moreover, the unbalanced number of JAKs (four) versus the cytokine pathways they transduce (over fifty) raises the question about the mechanisms that allow the versatile function of JAKs. For example, JAK1 is employed by the IL-2, IL-4, IL-10, and gp130 (including e.g., IL-6) receptor families as well as type I and type II interferons (e.g., IFNα and IFNγ, respectively). Intrigued by these questions, we set to study the role of JAK1 and particularly its JH2 in various signaling pathways. Our results show that JAK1 has varying roles in different receptor complexes, and that JH2 mediates important allosteric regulation of the JAK activity.

## 2. Results

### 2.1. JAK1 Is Dominant STAT Activator in IL-2, but not in IFNγ and IFNα Signaling

To investigate the role of individual JAKs in heterodimeric receptor complexes, we focused on IL-2, IFNγ, and IFNα receptor systems. These pathways utilize JAK1 but consists of different JAK dimers: IL-2-signaling is driven by JAK1-JAK3, IFNγ by JAK1-JAK2, and IFNα by JAK1-TYK2. First, we studied how dependent the signaling systems are to the presence of an individual JAK. For this, we used cell lines that lack specific JAK expression. JAK1 and JAK3 deficient U4Cγβ-, JAK2 deficient γ2A- or TYK2 deficient 11.1 human fibroblast cells were transiently transfected with one or both JAKs relevant to the pathway, and the expression of JAKs and STATs was detected from the cell lysates. In addition, the phosphorylation of STAT1 (pSTAT1) and STAT5 (pSTAT5) was assayed. Of note, HA-STAT5 was transfected into the U4Cγβ cells, since endogenous STAT5 could not be detected with this method. 

All of the studied cytokine receptor pathways required the expression of two different wild-type (WT) JAKs for the cytokine-dependent activation of STAT (Appendix A). In IFNγ and IFNα systems, STAT1 activation required the expression of both JAK1 and JAK2 or TYK2, respectively. In the IL-2 receptor complex both JAK1 and JAK3 were also required for cytokine dependent signaling; JAK1 alone activated STAT5 but the activation was unresponsive to cytokine while JAK3 could not induce STAT5 phosphorylation (even with IL-2) without the presence of JAK1 (Appendix A). Together these results confirm the previous findings that cytokine induced signaling requires heterodimerization of JAK WT [26,27] as well as validated the experimental system. 

Next, we studied the JAK dependency of the three signaling systems in the context of hyperactive mutations. JAK2 V617F mutant or homologous mutations in JAK1 and TYK2 (see Table 1) were transfected in JAK1 deficient cells either alone or with the relevant partner JAK. Interestingly, we observed that unlike in the WT-setting, JAK2 V617F and homologous mutations in JAK1 and TYK2 could induce (reduced) pSTAT1 even in the absence of the partnering JAK. However, the presence of the partner JAK increased the activation (Figure 1A). JAK1 V658F also activated STAT5 independently of JAK3 in the IL-2 system (Figure 1A [9]). As JAK1, JAK2 and TYK2 are roughly same size (~130 kDa) they appear as a single band in the western blot (with less intensive band when only one JAK is transfected with the vector).

In contrast, the pathogenic JAK3 R657Q residing in the JH2-JH1 interface was strongly dependent on the presence of JAK1 WT, shown as abolished STAT5 activation when JAK1 was not transfected into the U4Cγβ cells (Figure 1B [9]). Of note, equal amounts of DNA for each JAK were transfected although the JAK1 HA-signal is consistently weaker than JAK3 HA. As we have previously noticed that overexpression of JAK1 quickly increases the activation status of the basal STAT5 [9], and to keep the experimental set-up reliable, we did not want to increase the expression of JAK1 despite the weaker bands obtained with the immune labeling. In addition, the proper expression of JAK1 is evident also from the IL-2 responsive STAT5 activation, which cannot be induced without JAK1.

JAK3 R657Q was chosen as a representative activating JAK3 JH2 mutation because JAK3 lacks the residue homologous to JAK2 V617 that is present in other JAK JH2s (see Table 1). Furthermore, JAK3 has a leucine in the JH2 αC that in all other JAKs is occupied by a phenylalanine (Phe) (Table 1). This Phe is part of the so-called FFV-triad (Phe-Phe-Val) that is important for JAK2 V617F activation. The residue stack with the V617F, rigidify the αC and interacts with the SH2-JH2 linker via (JAK2) Phe 575, as well as alters the ATP-site cleft, leading to hyperactivation [28,29,30]. We were interested whether the difference in JAK3 FFV-triad (or the usage of non-homologous activating mutation) is causing the lack of STAT-activation in the absence of JAK1. Hence, we reconstituted the FFV-triad by double-mutating JAK3 at M592F (mimicking the JAK2 V617F) and at L570F to introduce the phenylalanine present in JAK1 WT (Phe F636) and JAK2 (Phe 595) (Figure 1C,D). The JAK3 M592F + L570F caused constitutive activation of STAT5 in the presence of JAK1 WT, but not in the absence of JAK1 WT. The single mutants were also analyzed for their STAT5-activation potential, and while JAK3 M592F showed some hyperactivation, L570F did not affect the STAT5 activation. The unaltered STAT activation in response to L570F was expected, as it simulates the WT-situation in JAK1 and JAK2. These results suggest that the dominance of JAK1 in the IL-2 system is not an intrinsic feature of JAK3, but rather a property of the IL2R complex.

Next, we compared the roles of JAK1 JH1 and JH2 in the IL-2 and IFNγ signaling. U4Cγβ cells were transiently transfected with JAK1 constructs where either JH1 or JH2 was deleted (Figure 1E). We have previously shown that in IL-2 signaling both JAK1 JH1 and JH2 are crucial for STAT5 activation (Figure 1E) [9]. However, in the IFNγ system JH1 deleted JAK1 maintained IFNγ responsiveness albeit the inductivity was considerably lowered, while JH2 was critical for any cytokine responsiveness (Figure 1E). This observation is in line with the study of Eletto et al., where JAK1 JH2, but not JH1, was found to be essential in IFNγ signaling, while both JH1 and JH2 were crucial for IFNα signaling [31]. 

Based on these results, the role of JAK1, and its JH2, is different between IL-2 and IFNγ (or IFNα) systems. In IL-2 signaling, JAK1 dominantly mediated the STAT5 phosphorylation and the activation requires both JH1 and JH2 domains. In the IFNγ and IFNα signaling, STAT1 activation requires both JAK1 and JAK2/TYK2, and IFNγ signaling shows to be dependent specifically on the pseudokinase domain of JAK1. 

### 2.2. L633K in the Outer Face of JAK1 JH2 aC-helix Inhibits WT and Hyperactive IL-2, IFNγ, and IFNα Signaling at Variable Degrees

We continued the analysis of JAK1 JH2 with available information from other JAK JH2s and focused on regions that have been identified to allosterically regulate JAK activation. We focused on the hydrophilic outer face of the JH2 αC and introduced a mutation in JAK1 that corresponds to JAK2 E592R (Figure 2A). JAK2 E592R inhibits hyperactivation in JAK2 and reduces V617F-driven dimerization [17]. 

αC-helix is an essential conserved region in protein kinases, and in the active kinase conformation the N-terminus of the αC-helix typically interacts with the activation loop phosphate. Furthermore, the conserved β3 Lys^72^ (numbering based on Protein Kinase A, PKA) couples the ATP phosphates to the αC-helix [25]. The clinical relevance of this region in pathogenic JAK1 signaling was evaluated by searching for patient-derived mutations that cluster in the area. Based on the COSMIC database [32] together with a literary research by Hammarén et al., the JAK1 JH2 αC-helix and the surrounding αB and β4 linkers were found to be highly mutated in human cancers. Based on mutations from Hammarén et al. [10], 11 out of the 29 residues (38%) were mutated and with all COSMIC mutations included, total of 16 mutations could be depicted in this region (Figure 2A, mutated residues shown as grey-shaded, bolded letters in the sequence). 

To test the effect of the αC modulation in cell-based assays, JAK1 L633K was transfected into U4Cγβ cells with STAT1 or STAT5 responsive luciferase reporters and the normalized luciferase values were compared to cells transfected with JAK1 WT. L633K reduced the cytokine responsiveness in JAK1-JAK3 driven IL-2 signaling and JAK1-TYK2 driven IFNα signaling (Figure 2B). In addition, the basal STAT5 activity in L633K transfected cells was reduced compared to WT. On the contrary, JAK1 L633K did not markedly affect the IFNγ responsiveness (driven by JAK1-JAK2), but slightly reduced the basal STAT1 activity. Basal STAT1 activity profile correlated with the reduced pSTAT1 in L633K transfected cells (Appendix A). Homologous mutations JAK2 E592R and TYK2 L653R also reduced the cytokine responsiveness as measured with STAT1 responsive IRF-GAS- and ISRE-luciferase reporters (Appendix A).

Since JAK1 L633K reduced IL-2-driven STAT activation, we were interested whether the mutation can inhibit constitutively activated JAKs in cis, as shown with JAK2 [17]. We created double mutants of JAK1 and JAK3, where the JAK1 L633K was combined with hyperactivating JAK1 V658F. In addition, similar mutations in JAK3 JH2 were tested (JAK3 E567R + R658Q, see Table 1). Both double mutants reduced the hyperactivation back to WT levels but retained the IL-2 inductivity (Figure 2C). Similar reduction of hyperactivation has been shown with homologous mutations in JAK2 [8,17] and were seen with TYK2 (Appendix A). Of note, more fluctuation in the STAT activation levels was observed in the JAK1 L633K + V658F double mutant, shown as larger errors seen in Figure 2C. However, the reduction from JAK1 V658F transfected cells was significant (*p* < 0.001).

### 2.3. Characterization of ATP Binding to JAK1 JH2

Next, we set to compare the inhibitory potential between the αC-mutant and another allosteric region of JH2, namely the ATP-binding site. First, we showed that in addition to JAK2 I559F and JAK3 I535F mutations that have previously been shown to inhibit ATP binding and JAK hyperactivation, [8,9] also homologous TYK2 V603F inhibits hyperactive TYK2 V678F in the IFNα system (Table 1, Appendix A). The mutation was originally designed to create steric hindrance in the pocket and have been veritably shown to inhibit ATP binding into JAK2 JH2 [8]. We introduced a mutation in JAK1 JH2 ATP-site, JAK1 I597F, which is homologous to the above-mentioned JAK mutants. In addition, another ATP site mutant, JAK1 K622A was chosen as its homolog has been shown to inhibit JAK2 and JAK3 hyperactivation in cis [8,9]. This highly conserved lysine (Lys72 in PKA) is critical in making a salt bridge to the conserved Glu (91 in PKA) in the αC, and is required for coordinating nucleotide binding of multiple kinases and pseudokinases [33]. 

We have previously noted that JAK1 I597F is unable to inhibit hyperactive IL-2 signaling, contrasting the effect of the homologous mutants in JAK2 and JAK3 [8,9]. Here, we found that JAK1 I597F increased basal STAT5 activity and pSTAT5 in WT background, although to a lesser extent than hyperactive JAK1 and JAK3 mutants (Figure 3A,B). Furthermore, the IL-2 induction was disturbed in comparison to JAK1 WT, and although some increase was apparent in the STAT5 transcriptional activity assay, JAK1 I597F could not significantly respond to IL-2 addition (*p* = 0.12 between the basal vs. IL-2, 50 ng/mL). The pSTAT5 analysis of the mutant showed more variability, but also in this setting both the increased basal activity and the disturbed cytokine responsiveness were detected (Figure 3A,B). Mutation of the conserved lysine K622 in the JAK1 JH2 ATP-binding site (Table 1) to alanine reduced the cytokine induced STAT activation, thus correlating with the behavior of the JAK2 [8] and JAK3 homologs (Figure 3B). 

To decipher the cause for the untypical behavior of the JAK1 JH2 ATP-site mutant I597F, we produced and purified JAK1 JH2s with I597F or K622A mutation. GST-tagged proteins were bound to glutathione sepharose, eluted by digestion with Tobacco Etch Virus TEV Protease and further purified in size-exclusion chromatography (SEC) (Figure 3C). The JAK1 JH2 mutants were then analyzed in differential scanning fluorimetry (DSF) with and without ATP. JAK1, JAK2, and TYK2 JH2s have been shown to bind ATP in the presence of divalent cations while JAK3 JH2 binds ATP without cations [8,9] and thus proteins were analyzed with and without MgCl_2_. Figure 3C shows the melting temperatures (Tm), as well as the change in the melting temperatures (dTm) relative to the WT JH2 apo-form (protein that does not bind any ligands). As expected, ATP binding did not increase the Tm in K622A. However, the mutant showed significantly increased thermal stability. Compared to the WT JH2 Tm (44 °C), the I597F mutant showed reduced Tm of 40 °C, and addition of 1 mM ATP and 1 mM MgCl2 increased the melting temperature by 2 °C. The increase was only slightly differed from the observed 3 °C increase in WT JH1. The Tm for apo-form WT, I597F, and K622A were 44.3 °C, 40.3 °C, and 53.3 °C, respectively.

Taken together, our cell-based and recombinant protein assays suggest that the JAK1 JH2 ATP-site differs from other JAKs in that the I597F cannot efficiently block ATP binding and shows increased basal activity and no inhibition with hyperactivating mutants. These unexpected effects to the signaling may be due to reduced stability of the mutated JH2 with I597F. This hypothesis is supported by data from JAK1 K622A that shows increased thermal stability and reduced signaling, probably stemming from the stabilizing interaction towards the αC that locks the whole domain in an inhibitory position.

### 2.4. JAK1 L633K Inhibits Hyperactive JAK3 but Does not Inhibit Hyperactive JAK2 and TYK2

To further depict the role of JAK1, and its JH2 in IL-2 and IFNγ pathways, we studied the inhibition potential of the JH2 mutations towards the partner JAK (inhibition in trans). The JAK1 L633K was first transfected with active JAK3 R657Q into U4Cγβ cells and the pSTAT5 as well as the activity of endogenous STAT5 were detected (Figure 4A, Appendix A). Correlating with the results presented above, JAK1 L633K had a strong inhibitory effect toward JAK3, while in the opposite experimental layout the αC mutated JAK3 E567R was unable to reduce JAK1 V658F hyperactivation. In the IFNγ system, however, JAK1 L633K was unable to effectively reduce JAK2 V617F hyperactivation (Figure 4B, Appendix A). This observation supports the previous results showing that JAK1 mutants have less effect in the IFNγ-driven STAT1 activation when compared to the IL-2 induced STAT5 activation (see Figure 2B). Reduction of the IFNγ signaling occurred both in basal and in IFNγ stimulated cells when the JAK2 αC-mutant E592R was co-transfected with activating JAK1 V658F. However, the variation in the STAT1 activity was large, and the JAK2 E592R driven inhibition was not significant (two-tailed *p*-value 0.06). Similarly, large variation was detected also in IL-2 system with homologous JAK1 mutant although in the IL-2 system a distinct inhibition was observed (Figure 4C, Appendix A). 

In the IFNα system, JAK1 L633K did not show inhibition of the co-transfected TYK2 V678F (Appendix A), and vice versa, TYK2 L653R at the JH2 αC did not show inhibition in the JAK1 V658F-driven basal pSTAT1 (Appendix A). However, TYK2 L653R reduced the IFNα-induced activation (Appendix A). In Figure 2B JAK1 L633K was previously shown to affect the IFNα-induced activation but not the basal STAT1 activation. This could indicate that TYK2 and JAK1 JH2 αC-helices are important for cytokine induced STAT1 activation, but not necessarily in maintaining the low basal activation state in the IFNαR complex.

### 2.5. JH2 Mutations in JAK Heterodimer Partners Show a Cumulative Inhibitory Effect

After establishing that JAK1 L633K in the αC strongly inhibits IL-2 signaling, and that JAK1 K622A at the ATP site also reduced hyperactivation in cis, we wanted to compare the inhibition potential between the two regulatory JH2 sites in trans. JAK1 JH2 ATP site mutant K622A reduced but did not abrogate hyperactivation in the JAK3 R657Q background (Figure 5A) and the mutation was similarly non-efficient against the JAK2 V617F driven STAT1 activation. In the work of Hammarén et al., the JAK2 JH2 ATP-site mutant I559F was found to lower the kinase reaction catalysis rates (k_cat_) [17], which could explain the reduction in the pSTAT-values shown also with JAK1 K622A. However, the results indicate that ATP mutants have lesser inhibitory effect in comparison to the effect driven by the αC mutants.

Modulating JAK JH2s can reduce constitutively active JAK signaling, but unlike the kinase-dead mutations that target the JH1 active site and completely abolish the signaling, JH2 mutants tend to maintain the cytokine responsiveness [8,9,17]. Thus, we were interested to see the cumulative effect of the JH2 mutants and transfected JAK1 and JAK3, both carrying a homologous αC mutation into U4Cγβ cells (Figure 5B). IL-2 titration shows that the STAT5 activity is abrogated when the αC-helix of both JAKs is mutated, and the same is true regarding the pSTAT5 status. Interestingly, when both of the JAK1 and JAK3 JH2 ATP-binding sites are mutated, the IL-2 response is severely diminished but the cells maintain their responsiveness to the cytokine (Figure 5B, right side). 

To be noted, the expression levels of the JAK3 mutants are slightly reduced in comparison to the JAK3 WT in the Figure 5B western blot, although this kind of variation in the expression levels was not generally observed between the JAK3 mutants. Moreover, the expression levels between the αC and ATP mutants are similar, and the results further supported by the transcriptional activity assay (luciferase assay above the western blot). Thus, the results can be considered representative in showing that the αC mutant combination abolishes IL-2 signaling, while ATP mutant maintains a IL-2 inductivity, albeit reduces the STAT activation. 

As suggested by Hammarén et al., the disruption of the dimerization interaction between JH2 domains can be mediated by the in the αC and mutants in this region may result in complete loss of signaling. The JH2 ATP-binding site, on the other hand, likely affects the kinase activity in cis via regulating JH1 and thus have a weaker inhibitory potential [17].

## 3. Discussion

JAK1 is the most widely employed JAK by a variety of cytokine receptors but the underlying molecular mechanisms of JAK1 regulation are still largely elusive. One interesting question is whether JAK1 function is conserved between the different receptor systems, and in this work, we investigated the role of JAK1 in IL-2, IFNγ, and IFNα signaling. 

JAK1 null mice die perinatally and the newborn mice display a strong reduction of thymocytes, highlighting its importance in immunological and hematological functions [34]. Accordingly, somatic JAK1 GOF mutations are found in 10–20% of T-cell acute lymphoblastic leukemia (T-ALL) patients [35]. Recently, conditional JAK1 knock-out (KO) mice were developed, and RNA sequencing from isolated hematopoietic stem cells showed that the genes most affected by the loss of JAK1 were STAT1, STAT2, and multiple members of the IFN regulatory transcription family, confirming the important role of JAK1 in regulating these factors [36]. Moreover, it was shown that loss of JAK1 leads to decreased IFNγ sensitivity ex vivo. Interestingly, constitutive JAK2 (V617F) activation could not rescue the defects in the JAK1 KO stem cells, which is supporting our observation of non-redundant functions of JAK1 and JAK2 in IFNγ signaling. 

The focus of our study was on the JAK1 pseudokinase (JH2) domain, where we compared its function in IL-2, IFNγ, and IFNα signaling. Our results show that in the tetrameric IFNγR-complex both JAK1 and JAK2 are required for STAT1 activation while in IL-2 signaling JAK1 functions plays a dominant role in STAT phosphorylation. Interestingly, in IL-2 signaling both JH1 and JH2 in JAK1 were indispensable for signaling while the JH2 domain in JAK1 was crucial for IFNγ induced STAT1 activation, and deletion of JH1 only reduced stimulation (Figure 1). Thus, our data supports the model suggested by Briscoe et al. where JAK2 kinase activity is required predominantly for initiating signaling and possibly for the phosphorylation of STAT1, whereas JAK1 phosphorylates the IFNγR1 and recruits STAT [37]. Eletto et al. further suggest that JAK1 JH2 domain is required for interaction with JAK2 and conclude that this interaction, rather than JAK1 kinase activity, is mandatory for JAK2 activation after IFNγ stimulation leading to STAT1 phosphorylation [31]. Furthermore, our studies specify that the outer face of the αC-helix in JAK1 JH2 is important in allosteric regulation of the IFNγ (and more strongly the IL-2) signaling. However, evidence exists also for a dominant role of JAK1 in STAT activation at IFNγ, and kinase-dead JAK1 was found to abolish phosphorylation of JAK2 while kinase-dead JAK2 only lowered JAK1 phosphorylation (in both cases the pSTAT1 signal was abolished) [38]. In a recent study JAK1 JH2 αC mutations A639F, E637R, F636A, F575A were shown to almost completely abolish IFNγ signaling, while similar mutations in JAK2 only reduced the cytokine responsiveness [39]. However, also in this study, the above-mentioned JAK1 mutants had the potential to inhibit JAK3 in trans, in line with our observation. Lastly, Keil et al. have reported an important scaffold function of kinase-dead JAK2 in a mouse model of IFNγ and concluded that JAK1 is the main activator of STAT1 [40]. Our data shows a distinct difference between the dominance of JAK1 in IL-2 and IFNγ systems and indicate that JAK2 JH2 mutants have greater potential to alter the IFNγ system than JAK1. However, we cannot unequivocally point JAK2 as a dominant similarly to JAK1 in IL-2. In conclusion, the interplay between heteromeric receptors and heteromeric JAK pairs involves intricate and receptor specific regulation.

Although the structures of the JAK1, JAK2, and TYK2 FERM-SH2 domains have been solved with erythropoietin receptor (EPOR), leptin receptor (LEPR), and interferon receptors [41,42,43], the dimerization mechanism of JAKs and their cognate receptors is not yet fully resolved. In both JAK2 structures, FERM-FERM interactions between JAK2-molecules were apparent, but the residues contributing them varied slightly in EPOR and LEPR [44]. In addition to the FERM-FERM dimerization, also a JH2-JH2 interaction has been suggested [45]. A plausible model for the activation of JAKs is loosening of the JH1-JH2 interface that opens the conformation of the full-length JAK, allowing transphosphorylation of the adjacent JH1s [45]. The model is supported by the electron microscope images of the full-length JAK1 where “closed” (inactive) and “open” (active) conformations were observed [46]. More direct evidence of the JH2 dimerization came from the study of Hammarén et al., who showed that mutating the JAK2 JH2 αC reduces constitutive receptor dimerization driven by the V617F mutation [17]. Since the JAK1 αC-mutant used in this article is homologous to the JAK2 E592R studied by Hammarén et al., we hypothesize that also in JAK1 the mutant reduces dimerization of the JAK JH2s and thus the receptors. Together, these observations allowed us to suggest a model where JAK1 drives the oligomerization of IFNγ-receptor (IFNγR) complex, while JAK2 is the initiator and main contributor in the activation of STAT1 (Figure 6). 

Individual JAKs show high specificity to distinct cytokine receptors. However, the study of Koppikar et al. showed that under some circumstances, conserved JAK pairing can be circumvented. They observed that long-term treatment with ruxolitinib results in resistance of JAK2 hyperactivation (persistence) that is caused by transphophorylation of JAK2 by JAK1 or TYK2 which allows constitutive STAT5 activation typically driven by homodimeric JAK2 [47]. Ruxolitinib is type I inhibitor targeting the active conformation of the kinase. Interestingly, the persistence could be overcome by applying type II inhibitor, which binds to the inactive JAK and locks it in the unphosphorylated form [48]. 

Recently, Tvorogov et al. showed that ruxolitinib induces dose-dependent pJAK2, which can cause life-threatening cytokine-rebound syndrome (due to re-activation of STAT) when the drug is withdrawn [49]. Again, the effect was not apparent when the JAK2 V617F expressing cells were treated with type II inhibitor. These studies show the importance to consider the trans-activation properties between JAKs as well as the active vs. inactive conformation of the protein, even if the ATP transferase is inhibited. Thus, the phosphorylation status of drug inhibited JAK2 appears to be critical for the development of persistence. In conclusion, in the JAK-receptor complex both kinetic and structural characteristics appear to be critical determinants in activation of JAK-STAT signaling. Future studies are required to depict the exact mechanism of the receptor complex activation (dimerization and phosphorylation). 

In line with previous studies in IL-2 signaling, we showed that JAK1 is dominant over JAK3, and JAK3 is incapable of inducing STAT activation in the absence of JAK1 [9,26]. Our data suggests that JAK3 does not directly phosphorylate STAT5 but is an important regulator of the cytokine responsive STAT5 activation, most likely through modulating the activation potential of JAK1. To study the roles of JAK3 and JAK1 more thoroughly, we reconstituted the JAK2 V617F homolog into JAK3 (M592F + L570F). Here, we showed that JAK2 V617F and homologous activating mutants in JAK1 and TYK2 could induce low levels of activation in the absence of the partner JAK, while in the WT context expression of both JAKs was crucial for functional signaling. However, the JAK1:JAK3 pair presents an exception as JAK1 WT could induce STAT5 activation in JAK3-deficient cells, although the activation was unresponsive to IL-2 (Figure 1). Interestingly, even with the V617F simulating, constitutively active JAK3 mutant M592F + L570F remained JAK1 dependent for STAT5 activation. 

The apo structure of JAK1 JH2 has been solved but its regulatory function has not been investigated in detail. We focused on the allosteric JAK1 JH2 regions and their role in different signaling pathways and found that mutating residue L633 in the solvent-exposed face of αC-helix inhibits JAK1-driven signaling. The L633K mutant resides in the regulative JH1-JH2 interface and recently, a homologous JAK2 E592R mutant was shown to inhibit constitutive signaling in EPOR and IFNγR systems [17,50]. The L633K mutation was most effective in inhibiting IL-2 signaling, in basal and cytokine-induces signaling. In addition, JAK1 L633K effectively reduced STAT activation in a background where JAK1 or JAK3 were constitutively active (inhibition both in cis and in trans).

Lastly, we introduced two mutations to disrupt ATP binding to JAK1 JH2. The JAK1 I597F mutation, homologs of which were shown to inhibit constitutive activation in JAK2 and JAK3 and TYK2, did not inhibit JAK1, JAK2, or JAK3 driven hyperactivation, but even increased the basal STAT5 activation. Interestingly, the mutation decreased the stability of the JH2 recombinant protein, and did not inhibit ATP binding. Another JAK1 JH2 ATP-site mutant, K622A inhibited STAT activation in both WT and JAK1 V658F context, and biochemical studies showed that the mutant stabilizes the JH2. To be noted, all tested JH2 mutants differed from the kinase-dead (JH1) mutants in that they maintained the cytokine inductivity. Combining the cell-based and recombinant protein derived data support mechanism of regulation where stabilizing JH2 inhibits hyperactivation while de-stabilizing further activates JAKs.

Taken together, we have obtained detailed information of the JAK1 JH2 and identified differences in JH2 function in different cytokine receptor pathways. These results underline the importance of thorough understand of the mechanism of JAK signaling as a means to create safer and more efficient inhibitors. 

## 4. Materials and Methods

### 4.1. Plasmid Constructs and Mutagenesis

Full-length human JAKs were previously cloned in pCIneo expression vector by using SalI-NotI restriction sites. Full-length human STAT5A was in pXM vector. JAKs and STAT5A were C-terminally HA tagged. Site-directed mutagenesis was performed with QuikChange (AgilentTechnologies, Santa Clara, CA, USA), according to the manufacturer’s instructions, and verified by using Sanger sequencing (see the primers that were used in Table 2). For luciferase reporter assays, firefly luciferase reporter constructs for STAT5 (SPI-Luc) or STAT1 (IRF-GAS/ISRE for IFNγ and IFNα respectively) were used together with a constitutively expressing renilla luciferase plasmid. 

### 4.2. Mammalian Cell Culture

JAK1 and JAK3 deficient U4Cγβ, JAK2-deficient γ2A, TYK2 deficient 11.1. human fibrosarcoma cells were cultured according to standard culturing conditions in Dulbecco modified Eagle medium (Lonza, Basel, Switzerland) supplemented with 10% FBS (Sigma, St Louis, MO, USA), 2 mmol/L L-glutamine (Lonza), and antibiotics (0.5% penicillin/streptomycin; Lonza). For transfection, cells were seeded on 24- or 96-well tissue-culture plates and transfected with FuGENE HD (Promega, Madison, Wisconsin, USA), according to the manufacturer’s instructions. Cells were transfected for 48 h and, where needed, cytokine stimulated in starvation medium without FBS for 15 min (for immunoblotting) or 5 h (for reporter assays), unless otherwise specified, with human recombinant IL-2 (PeproTech, Rocky Hill, NJ, USA), IFNγ (PeproTech) or IFNα (PeproTech).

### 4.3. Cell Transfection and Immunolabeling

Human fibroblast cell lines U4Cγβ, γ2A, and 11.1 (deficient in JAK1 and JAK3, JAK2 or TYK2, respectively) were transiently transfected using Fugene HD reagent with different human JAK-hemagglutinin (HA) constructs in pCIneo vector (75 ng for each per 24 well plate well). If two JAK constructs were transfected simultaneously, equal amounts of DNA were used. If STAT5 phosphorylation was studied, 2 ng or HA-tagged STAT5 was co-transfected. After 48 h transfection, cell were stimulated (if needed) for 20 min and washed with cold phosphate buffered saline (PBS). Triton X-100 lysis buffer with protease and phosphatase inhibitors (2 mM vanadate, 1 mM phenylmethanesulfonyl fluoride, 8.3 µg/mL aprotinin, and 4.2 µg/mL pepstatin) was used to lyse the cells. Whole cell lysates were spun for 20 min at 16,000 g at 4 °C, and the resulting supernatants were run on 4–15% Mini-PROTEAN® TGX™ Precast Gels (BioRad, Irvine, CA, USA). Immunoblots were blocked with bovine serum albumin (BSA) and incubated with primary antibodies for HA Tag (1:2000, OAEA00009, Aviva Systems Biology, San Diego, CA, USA), phosphorylated STAT1 (pSTAT1; 1:1000, #7649, Lot1 Cell Signaling), STAT1 (1:1000, 610116, BD Biosciences, San Jose, CA, USA), or phosphorylated STAT5 (pSTAT5; Cell Signaling, #4322, Lot9), and with a mixture of goat anti-rabbit and goat anti-mouse DyLight secondary antibodies (both from Thermo Fisher Scientific, Waltham, MA, USA). Blots were scanned with an Odyssey CLx (LI-COR Biosciences, Lincoln, NE, USA), and immunoblot signals were quantified with Image Studio software (LI-COR Biosciences) by manually assigning bands and dividing the phosphorylation (pSTAT1 or pSTAT5) signal values with the expression (STAT1 or HA) signals.

### 4.4. Luciferase and Dual Luciferase Assays

STAT5 transcriptional activity was assessed by measuring the luciferase expression (SPI-Luc 2) driven by a STAT5 responsive promoter, as described previously [2]. For STAT1 transcription activity detection, either IRF-Gas or ISRE-Luc plasmids (30 ng) were transfected instead of SPI-Luc 2 (transfected with 15 ng plasmid). These are specific for IFNγ and IFNα stimulated activation of STAT1, respectively [2,51]. U4Cγβ, 11.1, or γ2A cells were transfected with indicated DNA constructs including the STAT-reporter and with 15 ng renilla plasmid (pRL-TK). The latter was co-transfected as an internal transfection control. Transient transfections were done in 96 well plates with FuGENE HD (Promega) according to manufactures instructions. Then, 42 h after transfection, cells were stimulated (in starvation media) or starved for 5 h after which luciferase assays were analyzed using the dual luciferase reporter assay system (Promega) according to manufactures instructions. Luciferase values were measured with EnVision 2104 Multilabel Reader (Perkin Elmer, Waltham, MA, USA). The results are presented as relative luciferase activity (arbitrary units: a.u.) corresponding to the firefly luciferase light emission values divided by renilla luciferase light emission values.

### 4.5. Recombinant Protein Production and Purification

JAK1 JH2 constructs spanning residues 561–852 of human JAK1, of the wild-type sequence or either I597F or K622A mutations were sub-cloned into pFastBac vector for expression as N-terminal glutathione-S-transferase (GST) fusion proteins with a tobacco etch virus (TEV) proteins. Constructs were expressed in High Five insect cells (Thermo Fisher Scientific) using the Bac-to-Bac baculovirus expression system (Invitrogen, Carlsbad, CA, USA) according to manufacturer’s instructions. After protein expression (10% P3 virus, 48 h, 27 °C), the cells were collected by centrifugation, resuspended in lysis buffer containing 50 mM Tris HCl pH 8.0, 10% Glycerol, 500 nM NaCl, 1 mM TCEP supplemented with phosphatase and protease inhibitors (100 mM sodium orthovanadate, 100 mM PMSF, 10 µg/mL pepstatin A), and lysed by applying four freeze-thaw cycles. The lysates were clarified by centrifugation and incubated 1.5 h with GSH-coupled resin beads (Protino Glutathione Agarose 4B). Beads were washed and the protein detached by digesting with a tobacco etch virus (TEV) protease (overnight at 4 °C). The flow-through was concentrated and run on a Superdex 200 gel filtration column equilibrated in final buffer (20 mM Tris pH 8.0, 500 mM NaCl, 10% glycerol, 4 mM DTT and 0.02% CHAPS). The eluted peak was concentrated and stored at −80 °C.

### 4.6. Differential Scanning Fluorometry (DSF)

Thermal-shift experiments were carried out in 96-well PCR plates in a final volume of 25 μL with the following reagent concentrations: 6x Sypro Orange (Molecular Probes, cat. no. S6551), 3 μM protein Ni-NTA eluate, 1 mM MgCl2, 20 mM Tris pH 8.0, 500 mM NaCl, 10% glycerol, 4 mM DTT and 0.02% CHAPS, and 1 mM ATP. Reactions were heated in a real-time CFX96 PCR cycler (Bio-Rad) at 1 °C per min from 4 °C to 95 °C with a fluorescence read every 1 °C. Fluorescence data were then normalized to represent unfolding curves, which were fitted to a Boltzmann sigmoidal equation with GraphPad Prism to obtain average Tm values with errors as SD.

## 5. Conclusions

Our studies show that JAK1 has varying roles in IL-2, IFNα, and IFNγ systems. Our results demonstrate that the pseudokinase domain (JH2) of JAK1 is an important regulatory region and modulating it can either down, or up-regulate JAK1-driven signaling. Specifically the outer face of the JH2 αC-helix was effective in reducing wild-type and constitutive STAT activation, although the potency dependent on the JAK and the related signaling complex. Furthermore, we highlighted biochemical and biological differences between the ATP-binding sites of JAK JH2s, which can be beneficial in design novel JAK modulators targeting outside the conserved kinase domain.

## Figures and Tables

**Figure 1 cancers-12-00078-f001:**
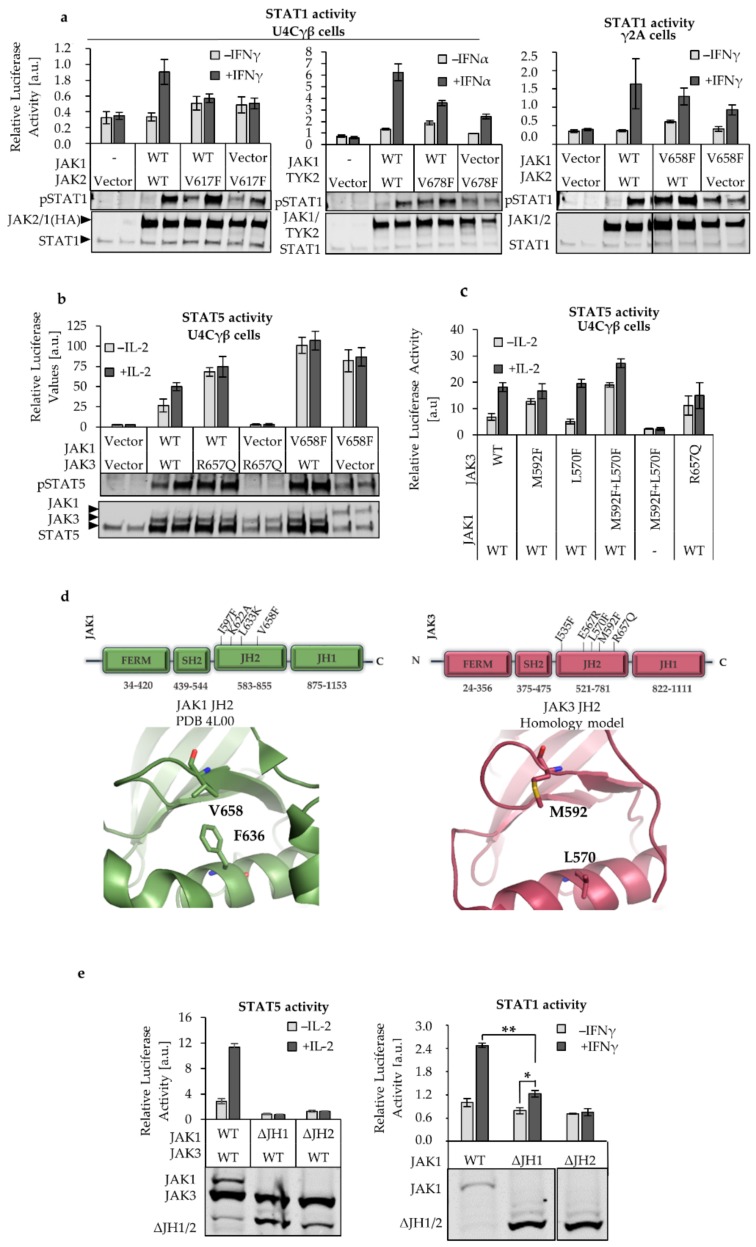
Comparison of JAK JH2s. (**a**) Janus kinases (JAK1, JAK2 and TYK2) hyperactive mutants can activate STAT1 without the partner JAK. pSTAT1 analysis and the transcriptional activity of STAT1 were detected from U4Cγβ and γ2A cells that were transfected for 24 h with JAK2 V617F, TYK2 V678F or JAK1 V658F. Wild-type (WT) partner JAK or vector was co-transfected with the activating mutants. Cells were starved overnight, and left untreated or stimulated with 100 ng/mL interferon γ (IFNγ) and interferon α (IFNα) for 20 min after which the pSTAT1 was detected by immunolabeling. For the STAT1 transcriptional activity detection, IRF-GAS or ISRE-luc plasmids) were co-transfected for 43 h with renilla plasmid (pRL-TK) (see Materials and Methods). Cells were stimulated or starved for 5 h, and the luciferase activity was measured. The values were divided with the renilla values to reduce the effects the possible differences in the transfection efficiency might have. Errors are SD of triplicates. Below are representative immunoblots of whole-cell lysates from U4Cγβ and γ2A cells transiently transfected with full-length hemagglutinin (HA)-tagged JAK mutants with or without JAK WT, as indicated. The cell lysates were immunolabeled with pSTAT1 (STAT1 Y701 phosphorylation), HA and STAT1 antibodies. The experiment was repeated twice with similar results. (**b**) JAK3 R657Q cannot induce STAT5 activation without JAK1. U4Cγβ cells were transfected with JAK WT or hyperactive JAK1 V658F or JAK3 R658Q and left untreated or stimulated with IL-2 (100 ng/mL) for 5 h. STAT5 specific SPI-Luc 2 plasmid was used for the detection of STAT5 transcriptional activity and the pRL-TK was used as a control. Errors are SD of triplicates. Below: Whole cell lysates from transiently transfected U4Cγβ cells were labelled with pSTAT5 (phosphorylation at Y694) and HA antibodies. HA-tagged STAT5 was transfected with JAK-HA constructs. Blot is a representative from three experiments. (**c**) Reconstituted JAK2 V617F homolog in JAK3 cannot signal without JAK1. JAK3 M592F, L570F, and double mutant were transiently transfected with JAK1 WT or vector. The U4Cγβ cells were starved and/or stimulated with 100 ng/mL IL-2, and the activity of the STAT5 responsive SPI-Luc vector measured as described above. (**d**) Illustration of JAK1 V658, F636 and homologous mutations in JAK3 pseudokinase domain (JH2) with schematic presentation of the four domains and approximate location of the JH2 mutations. Also the amino acid range for domains (according to the UniProt database) are shown in the scheme below each domain. Structures were visualized with PyMol using JAK1 JH2 structure (PDB 4L00) and JAK3 JH2 homology models (modelled based on TYK2 structure; Protein data bank 4OLI). (**e**) JAK1 JH2 is critical for IL-2 and IFNγ signaling. U4Cγβ cells were transfected with full-length JAK1 or with JAK1 JH1 or JH2 deletions. STAT5 and STAT1 responsive Luc-vectors were used as described before to detect the IL-2 and IFNγ responsiveness of the constructs. Errors are SD of triplicates, and *p*-values according to two-tailed student *t*-test (*—indicating *p* < 0.05 and **—*p* < 0.001). Expression of the HA-tagged, unstimulated JAK1 (and JAK3 in the IL-2 system) was confirmed by immunolabeling the whole cell lysates with HA-antibody. The band below the JAK1 WT and JAK3 bands in the left side panel WT/WT sample is due unspecific binding of the HA antibody.

**Figure 2 cancers-12-00078-f002:**
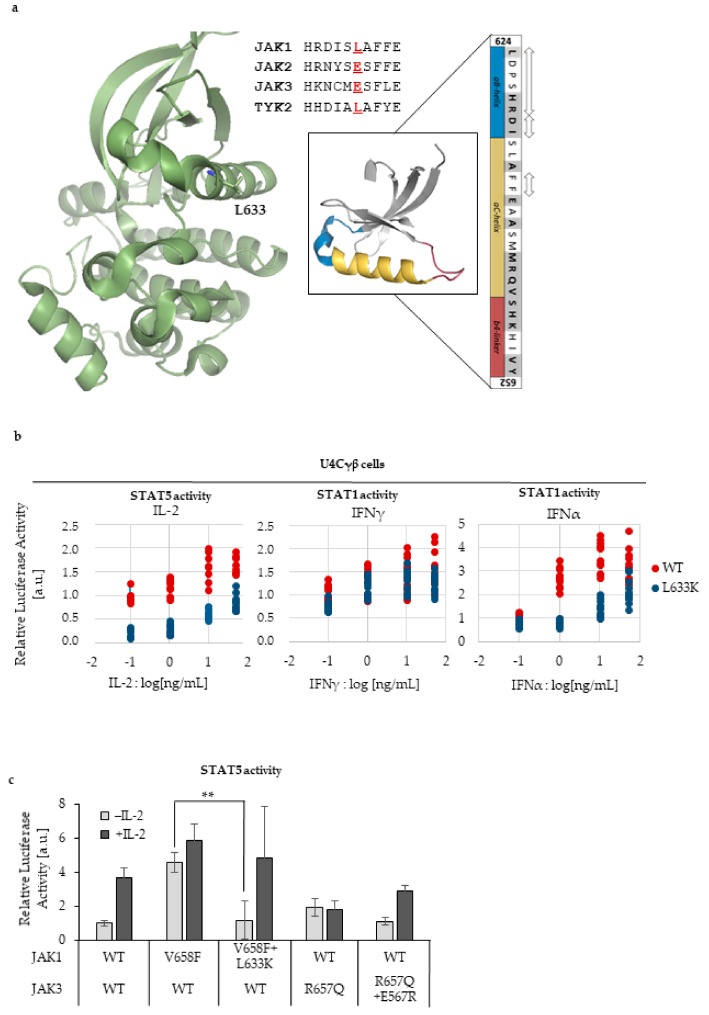
Analysis of the JAK1 JH2 αC-mutation. (**a**) JAK1 JH2 (PDB 4L00) structure is shown with the L633 in the αC-helix. The sequence around JAK1 L633 in JAK family is shown. **Right:** Mutations in the JAK1 αC-region including the adjacent αB-helix and the β4-linker. The mutations (shaded residues with bold letters) are derived from the review of Hammarén et al. 2018 [17] and the Catalogue of Somatic Mutations in Cancer (COSMIC)-database [32]. Deletions are shown as arrows. (**b**) Mutation in the JAK1 JH2 αC inhibits JAK1 driven IL-2 and IFNα signaling, but has lesser effect in the IFNγ-induced signaling. JAK1 WT (shown as blue dots) or JAK1 L633K (red dots) were transiently transfected with STAT1 and STAT5-responsive luciferase plasmids as described previously. The U4Cγβ cells were then starved or stimulated with increasing amounts of cytokine, and the STAT-activation was detected. All 12 replicas are presented as dots in a logarithmic axis showing the cytokine amount versus the relative luciferase activity. Basal state is set to −1. (**c**) JAK1 L633K and homologous mutation in JAK3 JH2 αC reduces hyperactivation in *cis*. JAK1 and JAK3 mutations were studied with (100 ng/mL) and without IL-2 stimulation in U4Cγβ cells that lack JAK1 and JAK3. In comparison to the activating mutations, the double mutants had reduced basal activity and responded to cytokine stimulation similarly as JAK WT transfected cells. STAT5 activity was studied with SPI-Luc luciferase system as described previously. The errors are SD of two separate experiments both having triplicate samples (*n* = 6). Two-tailed *t*-test was performed and **—indicates *p*-value <0.001.

**Figure 3 cancers-12-00078-f003:**
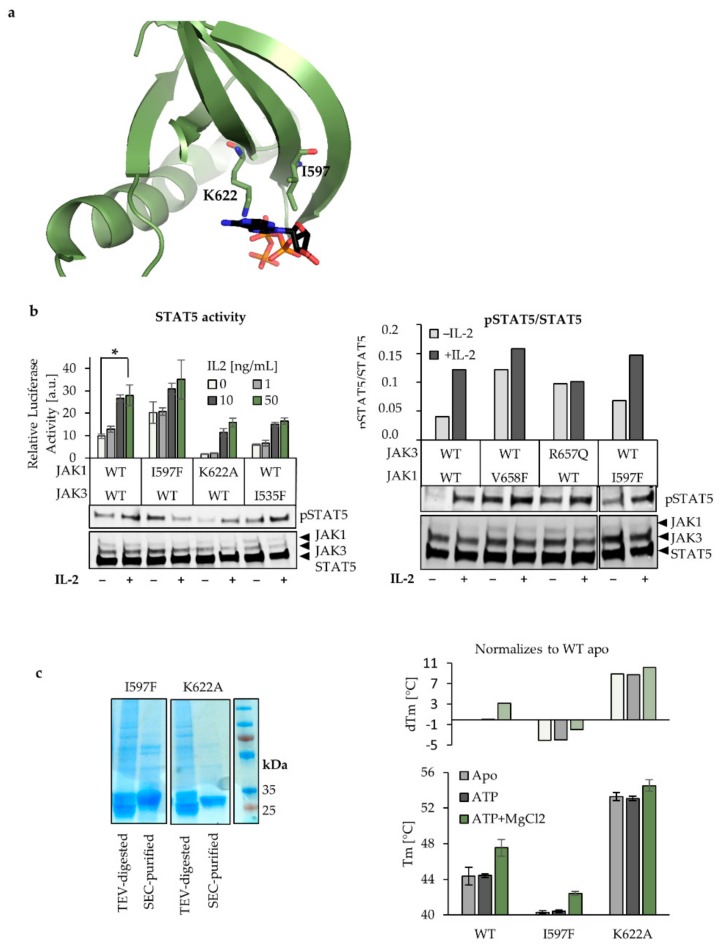
Characterization of the JAK1 JH2 ATP-binding site mutants. (**a**) Illustration of the JAK1 JH2 ATP-binding pocket, including the αC-helix of (PDB 4L00). The mutated residues K622 and I597 are shown, as well as ATP. (**b**) JAK1 I597F slightly increases the basal STAT5 activity and is responding to IL-2. JAK1 K622A shows reduced but cytokine-responsive STAT activation. STAT5 responsive luciferase system was used as previously described in U4Cγβ cells transfected with JAK1 and JAK3 JH2 ATP-site mutants or JAK WT. The errors are SD triplicate samples. **Below**: pSTAT5- and HA- labeled cell lysates from basal, and cytokine treated cells. Two-tailed *t*-test was performed and *p* < 0.05 is indicated as *. **Right**: comparison of JAK1 I597F with WT and hyperactive JAK1 and JAK3. Immunoblots from whole-cell lysates were labelled with HA (JAK1/JAK3/STAT5) and pSTAT5 antibodies to detect the pSTAT5/STAT5 ratios for basal and IL-2 stimulated (100 ng/mL) cells. (**c**) Recombinant JAK1 JH2 mutants vary in their stability. Differential scanning fluorimetry (DSF) analysis of size-exclusion chromatography (SEC)-purified JAK1 JH2 proteins show that JAK1 I597F has lower Tm compared to WT, while K622A increases Tm. **Left**: the SDS page gels with JAK1 JH2 elutions before- and after SEC purification. The size of the JAK1 JH2 is ~34 kDa. Right: DSF analysis showing protein Tm −/+ ATP and −/+ MgCl2. Errors are SD (*n* = 6). Graph above presents the dTm that is normalized to the wild-type JAK1 JH2 in its apo-form (protein that does not bind any ligands).

**Figure 4 cancers-12-00078-f004:**
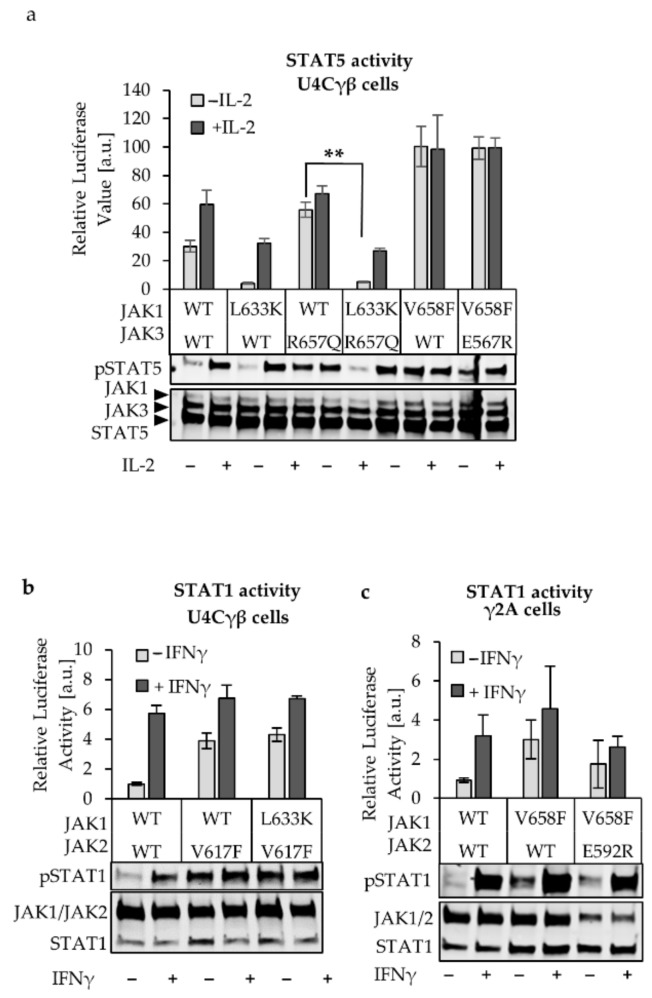
JAK1 L633K effect in trans. (**a**) JAK1 L633K reduces hyperactive JAK3 R657Q while homologous JAK3 E567R cannot reduce JAK1 V658F-driven activation. Both the transcriptional activity and the phosphorylation status of STAT5 were studied as previously described. Normalized SPI-Luc luciferase signal shows averages and SD of six experiments. HA- and pSTAT5 antibodies were used to detect the expression of JAKs and the STAT5/pSTAT5 ratio of non-stimulated and stimulated (100 ng/mL IL-2) cells. **—indicates *p* < 0.001 (two-tailed *t*-test) (**b**) JAK1 L633K does not efficiently inhibit JAK2 V617F driven STAT1 activation. JAK1 WT or L633K were transiently transfected with JAK2 V617F, and STAT1 (IFNγ) responsive IRF-GAS plasmid was used to detect the STAT1 activity as described earlier. Errors are SD of triplicate samples. Below: whole cell lysates from unstimulated (basal-state) cells transfected with JAK1 and JAK2 and labelled with HA and pSTAT1 antibodies. The blots are representative of three experiments. (**c**) JAK2 E592R inhibits JAK1 V658F-driven STAT1 activation. JAK2 WT or E592R were transiently transfected with JAK1 V658F and STAT1 (IFNγ) responsive IRF-GAS plasmid into γ2A cells that lack JAK2, and the STAT1 activity was detected as above. Errors are SD of six replicas. Below: whole cell lysates from basal-state and stimulated (100 ng/mL IFNγ) cells that were transfected with JAK1 and JAK2 and labelled with HA and pSTAT1 antibodies.

**Figure 5 cancers-12-00078-f005:**
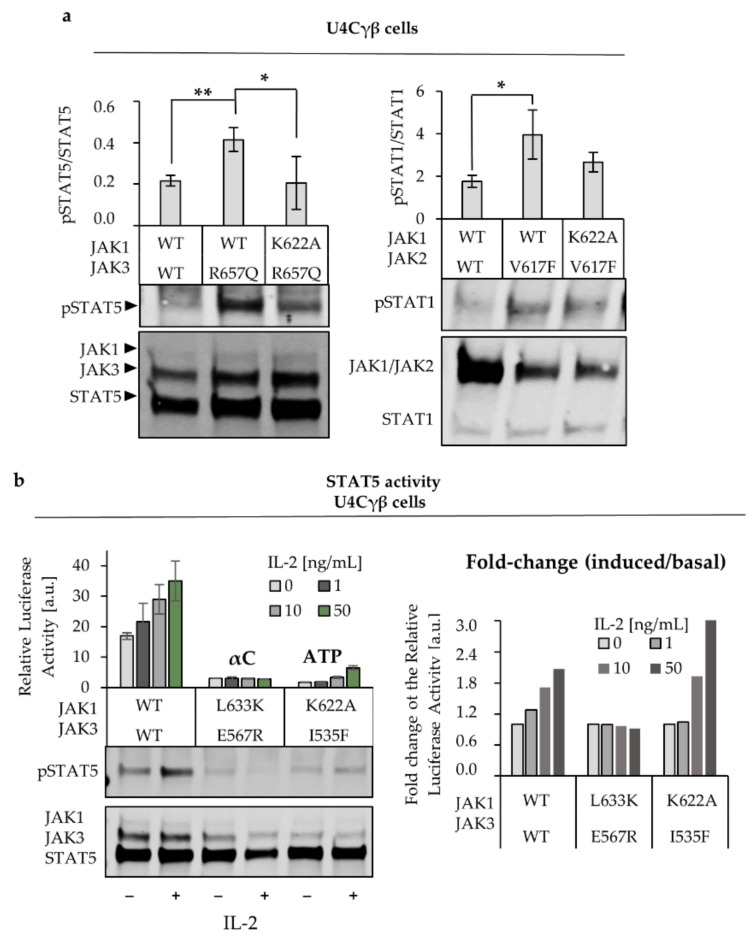
Characterizing the cumulative effect of the JAK JH2-mutants. (**a**) JAK1 K622A had variable reduction potential towards hyperactive JAK3 R657Q or JAK2 V617F. pSTAT1/5 was detected from cells transfected with HA-JAK1 and JAK3 or JAK2. The lysed cells were blotted to a membrane and labelled with HA, STAT1, and pSTAT1/5 antibodies as prescribed previously. Errors are SD of triplicate experiments. Two-tailed student *t*-test was performed between the WT and K622A in JAK3 R657Q background and indicates as * (*p* < 0.05) or ** (*p* < 0.001). (**b**) JAK1 and JAK3 with homologous JH2 αC mutations were co-transfected into U4Cγβ cells, which show abolishes IL-2 signaling. Similar setting with JAK1 and JAK3 ATP-site mutants can weakly respond to stimulation with 100 ng/mL of IL-2. STAT5-responsive plasmid was transfected as previously with JAK1 and JAK3 αC and ATP site mutants and after 43 h an increasing amount of cytokine was added into cells or the cells were only starved. Relative luciferase values of triplicates were detected and error are SD. **Below**: pSTAT5 analysis of cells transfected with JAK1 and JAK3 and treated, or not, with 50 ng/mL of IL-2. **Right**: the values shown in left were normalized to the basal values for each mutant (and WT) pair to show the fold change between the unstimulated and stimulated cells.

**Figure 6 cancers-12-00078-f006:**
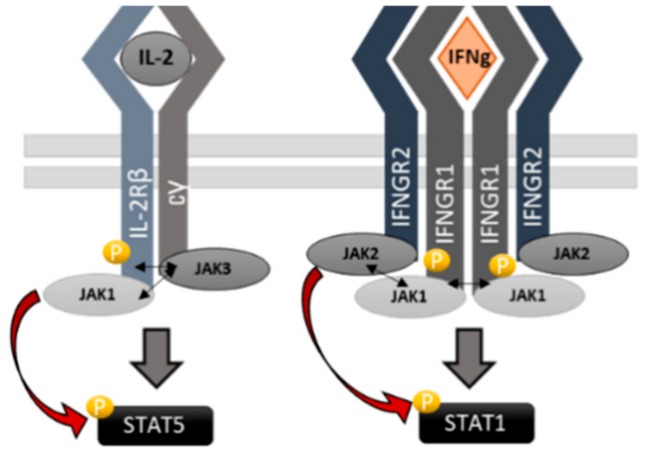
Illustration of the suggested activation cascades in IL2R and IFNγR systems. **Left**: Binding of IL-2 induces dimerization of the receptor subunits, possibly via JAK1 and JAK3 JH2-JH2 and FERM-FERM interaction allowing JAK1/3 transphosphorylation and activation leading to activation of STAT5 by JAK1. **Right**: In IFNγR JAK2 mediates STAT1 phosphorylation while JAK1 contributes by phosphorylating the IFNGR1, hence creating a docking site for the STAT1 and strengthening the oligomerized complex conformation.

**Table 1 cancers-12-00078-t001:** Mutations used in this study qualified as loss-of-function mutations (LOFs) or gain-of-function mutations (GOFs) based on the shown effects (-, designates as neutral).

JAK	Mutation	Effect	Short Description.
JAK1	L633K	LOF	At the solvent exposed face of the JH2 αC-helix, homologous to the JAK2 E592R.
	I597F	GOF/-	Residing JH2 ATP-binding site and designed to inhibit ATP binding. Homologous to JAK2 I559F.
	K622A	LOF	Removes conserved β3 lysine in JH2. Designed to inhibit ATP binding. Homologous mutations shown to inhibit hyperactivation in JAK2 and JAK3.
	V658F	GOF	Homologous to JAK2 V617F and TYK2 V678FF. Resides in the JH2 β4-β5 loop and potentially disturbs the SH2-JH2 linker and causes cytokine independent activation. Mutation in JAK1 or JAK2 cause ALL.
JAK2	E592R	LOF	At the solvent exposed face of the JH2 αC-helix, shown to inhibit JAK2 V617F-driven dimerization of EPOR [19].
	I559F	LOF	In the JH2 β2. Designed to sterically inhibit ATP binding and shown to inhibit ATP binding in recombinant JH2 [8].
	V617F	GOF	Homologous to JAK1 V658F and TYK2 V678F.
TYK2	L653R	LOF	At the solvent exposed face of the JH2 αC-helix, homologous to the JAK2 E592R.
	V603F	LOF	At the ATP, binding pocket of JH2, designed to inhibit ATP binding. Homologous to JAK2 I559F.
	V678F	GOF	Constitutive active mutation in JH2. Homologous to JAK1 V658F and JAK1 V617F.
JAK3	E567R	LOF	Resides at the solvent exposed face of the JH2 αC-helix. Homologous to the JAK2 E592R.
	I535F	LOF	At the ATP binding pocket of JH2 and homologous to JAK2 I559F.Shown to inhibit constitutive JAK3 activation [9].
	R657Q	GOF	Activating mutation found in ALL patient. Resides in the JH1-JH2 interface.
	M592F	GOF	Homologous to constitutively active JAK1 V658F, JAK2 V617F and TYK2 V678F.
	L570F	-	Mutation designed to create the WT state as in JAK1, JAK2 and TYK2 (F595 in JAK2). Stacks with the mutated JAK2 V617F and enables the hyperactivation via the FFV-triad formation (see text).
	M592F + L570F	GOF	Double mutant designed to create a complete FFV-triad into JAK3 (see above).

**Table 2 cancers-12-00078-t002:** Primer sequences for mutations used in this study:

	Fw	Rev
**JAK1ΔJH2 (583–855)**	atcctcaagaaggatctgaaaccagcaactgaagtggacccc	cttcagttgctggtttcagatccttcttgaggatccgatcg
**JAK1Δ JH1-HA (583–1153)**	ctgaaaccagcaactgaagtgtacccatacgatgttccagattacgcttag	ctaagcgtaatctggaacatcgtatgggtattcagttgctggtttcagatccttctt
**JAK1 L633K**	cagggatatttccaaggccttcttcgaggc	gcctcgaagaaggccttggaaatatccctg
**JAK1 I597F**	gagaacacacttctattctgggaccctgatgg	cccagaatagaagtgtgttctcgtgcctctcc
**JAK1 V658F**	ctatggcgtctgtttccgcgacgtggag	ctccacgtcgcggaaacagacgccatag
**JAK1 K622A**	gaagataaaagtgatcctcgcagtcttagaccccagccacagg	cctgtggctggggtctaagactgcgaggatcacttttatcttc
**JAK2 E592R**	gcacacagaaactattcacggtctttctttgaagcagc	gctgcttcaaagaaagaccgtgaatagtttctgtgtgc
**JAK2 V617F**	atggagtatgtttctgtggagacgagaatattctgg	tcgtctccacagaaacatactccataatttaaaacc
**JAK2 I559F**	ggccaaggcacttttacaaagttttttaaaggcgtacgaagagaagtagg	cctacttctcttcgtacgcctttaaaaaactttgtaaaagtgccttggcc
**TYK2 V603F**	cacaaggaccaacttctatgagggccgcc	ggcggccctcatagaagttggtccttgtg
**TYK2 L653R**	ccatgacatcgcccgggccttctacgagacagccagcc	cgtagaaggcccgggcgatgtcatggtgactagggtcc
**TYK2 V678F**	gcatggcgtctgtttccgcggccctga	tcagggccgcggaaacagacgccatgc
**JAK3 I535F**	ggtccttcaccaagttttaccggggctgtcgc	gcgacagccccggtaaaacttggtgaaggacc
**JAK3 L570F**	ggagtcattctttgaagcagcgagcttgatgagcc	ctcgctgcttcaaagaatgactccatgcagttcttgtgc
**JAK3 M592F**	ggcgtgtgctttgctggagacagcaccatggtgcagg	gtctccagcaaagcacacgccgtggagcagcacgagatgccgg
**JAK3 E567R**	gcacaagaactgcatgcgttcattcctggaagc	gcttccaggaatgaacgcatgcagttcttgtgc
**JAK3 R657Q**	aaggtgctcctggctcaggagggggctgatggg	cccatcagccccctcctgagccaggagcacctt

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
