# Peer review of "Characterization of JAK1 Pseudokinase Domain in Cytokine Signaling"

_cancers, 2019, doi:10.3390/cancers12010078_

Round 1
Reviewer 1 Report
The manuscript presented Raivola et al addresses a very important phenomenon, that of JAK heterodimerization. Understanding the mechanism of this heterodimerization has become particularly important after it has been shown that accumulation of JAK2 phosphorylation and heterodimerization with other JAKs (JAK1, JAK3 andTyk2) is the most possible mechanism of Ruxolitinib resistance in myelofibrosis (Koppikar et al Nature 2012). From the point of view of “signalling” the mechanisms underlying heterodimerization and trans-activation of JAKs are still largely elusive and the current manuscript utilizing three different cytokines (IL-2, INFγ and INFα), corresponding membrane receptors and four different JAK molecules, ads significant knowledge about it. The mutagenesis studies are well done and the authors demonstrate deep and detailed understanding of JH1/JH2 domain structure. The observation that JH2 rather than JH1 is required for INFγ-induced STAT1 activation is interesting and novel. However, although in the abstract and introduction the authors clearly state the clinical importance of JAK heterodimerization, the manuscript has very little connections with clinical observations and resistance to inhibitors such as Ruxolitinib. The absence of these links somewhat reduces the impact of the paper.
Major points:
The authors have to mention and discuss phosphoJAK2 accumulation/heterodimerization as a possible mechanism of resistance to JAK inhibitors, as raised in previous papers (Koppikar et al Nature 2012 and Tvorogov et al, Science Advances 2018). The authors have utilized JAK1V658F mutant as a form of hyperactivated JAK1 which can increase activation of other JAKs. A recent publication by Tvorogov et al, Science Advances 2018 has shown that JAK1 kinase activity may contribute to Ruxolitinib-induced accumulation of phosphor-JAK2. The authors should examine V658F mutation in JAK1 for its effect on the kinetics and magnitude of phosphor-JAK2 accumulation after Ruxolitinib treatment. This would improve the clinical relevance of these studies. The activation of signalling as shown in the manuscript is assed through phosphorylation of one STAT molecule (STAT1 or STAT5) and corresponding luciferase activity. This approach excludes receptors as substrates of JAK activity, thus signalling may still occur. The authors should add at least for Fig 1A, B, E, and Fig 3B IP/WB experiments examining the possible phosphorylation of the cognate receptor.
Minor points:
Table 1 for each mutation needs to be qualified as GOF or LOF to make it easy for the reader Suppl. Fig 1 is very hard to read. S1A misses TYK2. In the Discussion, cross-activation, heterodimerization and phosphor-JAK accumulation as a possible mechanism of resistance to JAK inhibitors should be discussed (and related to Koppikar et al Nature 2012; Tvorogov et al, Science Advances 2018).
Author Response
We thank the reviewer for a thorough and insightful review, and acknowledging the relevance of the study.
Major point: The reviewer raised an important matter of the link between mechanistic studies and clinical relevance and suggested to discuss Phospho-JAK2 accumulation/hetrodimerization as a possible mechanism of resistance to JAK inhibitors.
The following section has now been added to the Discussion:
Individual JAKs show high specificity to distinct cytokine receptors. However, the study of Koppikar et al. showed that under some circumstances, conserved JAK pairing can be circumvented. They observed that long-term treatment with ruxolitinib results in resistance of JAK2 hyperactivation (persistence) that is caused by transphophorylation of JAK2 by JAK1 or TYK2 which allows constitutive STAT5 activation typically driven by homodimeric JAK2 (Koppikar et al, Nature 2012).Ruxolitinib is type I inhibitor targeting the active conformation of the kinase. Interestingly, the persistence could be overcome by applying type II inhibitor, which binds to the inactive JAK and locks it in the unphosphorylated form (Levine et al. Cancer Cell, 2015).
Recently, Tvorogov et al. showed that ruxolitinib induces dose-dependent phospho-JAK2, which can cause life-threatening cytokine-rebound syndrome (due to re-activation of STAT) when the drug is withdrawn (Tvorogov et al, Science Advances 2018). Again, the effect was not apparent when the JAK2 V617F expressing cells were treated with type II inhibitor. These studies show the importance to consider the trans-activation properties between JAKs as well as the active vs. inactive conformation of the protein, even if the ATP transferase is inhibited. Thus, the phosphorylation status of drug inhibited JAK2 appears to be critical for the development of persistence. In conclusion, in the JAK-receptor complex both kinetic and structural characteristics appear to be critical determinants in activation of JAK-STAT signaling. Future studies are required to depict the exact mechanism of the receptor complex activation (dimerization and phosphorylation).
The reviewer also suggested to examine V658F mutation in JAK1 for its effect on the kinetics and magnitude of phosphor-JAK2 accumulation after Ruxolitinib treatment. This is an interesting and important aspect that also previous studies have addressed.
To see, if JAK1 can phosphorylate JAK2 in the presence of JAK2 inhibitor, Tvorogov et al. used g2A/IL3Ra/bc cells stably expressing wild-type or kinase-inactive JAK2, which were treated with JAK2 inhibitor (Fedratinib; 35-fold higher potency toward JAK2 than JAK1) to block specifically JAK2 phosphorylation. In their experiments, JAK1 phosphorylated JAK2 in the presence of JAK2 specific inhibitor.
Levine et al. (Cancer Cell, 2015) used g2A cells stably expressing constitutively active JAK1 V658F and kinase-dead JAK2 K882E and showed that type II inhibitor CHZ868 abrogated JAK1-mediated phosphorylation of JAK2 K882E at concentrations that did not fully inhibit JAK1 autophosphorylation.
Finally, Koppikar et al. observed a persistent JAK2 phosphorylation in JAK1V658F/JAK2K882E γ2A cells exposed to Ruxolitinib for longer times (4-24h) at concentrations sufficient to inhibit JAK2 autophosphorylation.
Our experimental system was developed on depicting the immediate activation mechanisms and as such not perfectly suited for the persistence studies. We think that previous studies have established well the role of JAK1V658F in phosphorylating JAK2 and refer to these studies in text.
The Reviewer also pointed out the phosphorylation state of the cytokine receptors and requested to add data in Figure 1A,B,E and Figure 3B by using IP/WB techniques.
We agree with the Reviewer that the receptor component in the present study indeed remains vague. The matter is examined in our laboratory in on-going imaging studies. Regarding the receptor phosphorylation studies unfortunately we were not able to obtain/find suitable functional antibodies for pIL-2Rg, pIL-2Rb or pIFNGR2 or produce tagged receptors within the resubmission timeline. Thus, we thank the reviewer for important suggestion and will hopefully be able to address the matter more thoroughly in our future studies.
Minor points:
As requested, the following addition to Table 1 has been added: each mutation is labelled as GOF or LOF (or neutral) based on the observed effect, and additional information is included to familiarize the reader better with the multitude of mutations. To clarify the Supplementary Figure 1, the figures have been modified, including the addition of TYK2 in S1A.
Reviewer 2 Report
The authors are attempting to gain mechanistic insight on how JAK1 interacts with different receptor complexes, which in turn results in activation of one cytokine over the other. The concept is interesting and will be useful for designing JAK inhibitors for clinical applications. However several major points need to be addressed and it is difficult to assess the manuscript till these are not fixed:
These include
Introduction:
1. Could benefit from a schematic of the domain in the four JAKs and the mutations that the authors are characterizing
Results:
1. The authors keep referring to Table 1, which only provides a list of mutations in the different JAKs, however, it does not highlight how these sites are analogous to each other, this needs to be clearly addressed
2. The result sections refers to incorrect figures either in the main text or the supplementary data or to data that is not found anywhere in the manuscript
3. The results need to be clearly described, it is very difficult to follow in the current format
Examples include:
“None of the cytokine induced signaling pathways 98 was functional without both wild-type (WT) JAKs expressed (Figure S1A,B).” Very vague statement. Do the author means can they please explain this result better. It is unclear which two JAKs they are refereeing to for which cell line and also which cytokine pathway are the results for?. Also Supplementary Figure 1a, the figure seems to have things cut out…?
“However, JAK1 alone 99 activated STAT5 in an IL-2-independent manner, while JAK3 could not induce STAT5 100 phosphorylation without the presence of JAK1.” Where is this result?
Results 2.1 “2.1. JAK1 is dominant STAT activator in IL-2-, but not in IFN and IFN signaling.” Is extremely difficult to follow. It is unclear from the text and the figures, what is the reporter molecule being tested, what is being added, there are no statistics etc. The authors need to clearly define the question, followed by how the experiment addresses the question and then describe the results, step by step. This section needs a major re-write.
“Based on mutations from Hammarén et al. [34], 11 out of the 29 residues (38 %) were mutated and 187 with all COSMIC mutations included, total of 16 mutations could be depicted in this region (Figure 188 1a).” - Figure 1a does not show the region - is this correct?
Other:
There are several typographical and grammatical errors throughout the manuscript
Author Response
We thank the Reviewer for the comments and we hope that our responses have improved the clarity of the manuscript. Below, are our answers to the points (marked 1.-4.).
Reviewer requested a schematic domain illustration with mutations that has been added to Figure 1D.
Reviewer mentioned that the mutation sites in Table 1 should be more carefully described. Information has been added to the Table 1, and each mutant has been labelled as LOF or GOF based on the seen effect (as suggested by Reviewer 1).
As requested by the Reviewer, we have corrected and clarified the result section, especially the section 1. Headings have been added to all luciferase graphs to clarify which effector (STAT1 or STAT5) is detected.
To the specific points raised by the reviewer, the answers are listed below (with the question):
Reviewer: “None of the cytokine induced signaling pathways 98 was functional without both wild-type (WT) JAKs expressed (Figure S1A,B).” Very vague statement. Do the author means can they please explain this result better. It is unclear which two JAKs they are refereeing to for which cell line and also which cytokine pathway are the results for?”
The chapter has now modified as follows:
All of the studied cytokine receptor pathways required the expression of two different wild-type (WT) JAKs for the cytokine-dependent activation of STAT (Figure S1A,B). In IFNγ and IFNα systems, STAT1 activation required the expression of both JAK1 and JAK2 or TYK2, respectively. In the IL-2 receptor complex both JAK1 and JAK3 were required for cytokine dependent signaling; JAK1 alone activated STAT5 but the activation was unresponsive to cytokine while JAK3 could not induce STAT5 phosphorylation (even with IL-2) without the presence of JAK1 (S1C).
Reviewer: Also Supplementary Figure 1a, the figure seems to have things cut out…? We did not observe any cut figures, but have cleared and revised the figure. Reviewer: “However, JAK1 alone 99 activated STAT5 in an IL-2-independent manner, while JAK3 could not induce STAT5 100 phosphorylation without the presence of JAK1.” Where is this result? Supplementary Figure S1C (reference added to the text). Reviewer: “Based on mutations from Hammarén et al. [34], 11 out of the 29 residues (38 %) were mutated and 187 with all COSMIC mutations included, total of 16 mutations could be depicted in this region (Figure 188 1a).” - Figure 1a does not show the region - is this correct? Reference corrected, to Figure 2A.
Reviewer 3 Report
The authors have contributed a significant body of work in the JAK field and are pioneers in addressing questions regarding allosteric regulation of JAK1 JH2, the consequence it has on JAK partners as well as downstream cytokine signaling and the therapeutic potential of targeting the JH2 domain vs. typical JH1 drugs. Clearly extensive studies were performed to provide compelling evidence toward addressing these challenging questions. This work might benefit from additional statistical analysis to bolster the claims and support the significance of the findings on allosteric regulation of the Jak1 JH2 domain in relation to cis- and trans- inhibition. Undoubtedly, the authors undertook a difficult task in reconstituting multiple cytokine signaling systems and in some figures the expression of Jak1 detected by Western blot is faintly detectable. Have the authors consider how this may limit the reader’s ability to interpret the consequence of JH2 JAK1 mutants in contrast to the amount of JAK1 expression? Lastly, as indicated below in specific comments, a few figures would benefit from added controls to help the readers draw conclusions from the trends seen in their extensive analysis.
Materials & Methods Section
Section 4.1 should include the primer sequences used to create all the mutations utilized in this study. Section 4.3, the quantities of DNA used for transfection (µg) should be included for each plasmid. Readers will be interested to know the quantities used (µg/transfection) from experiment to experiment. Section 4.3 the catalog number for pStat5 purchased from Cell Signaling should be included so that the reader can determine the species in which the antibody was generated.
Results Section
Figure 1a-b, would benefit from adding a WT/WT control for all three transfection systems shown. Readers will be able to draw a more powerful conclusion of how the absence of Jak partners affect cytokine signaling compared to WT/WT contorl. Figure 1a, 3rd Western blot panel for γ2A cells, is not clear if the JAK (HA) blot is for JAK1 or Jak2. If the blot in question is for Jak1, then the results are difficult to make conclusions, as there is not enough Jak1 (V658F) expression in the presence of vector only for comparison. If the blot in question is for Jak2, then labeling is needed to clarify as this panel is different from the 1st and 2nd in the series (U4Cγβ cells); additionally, a Jak1 blot would be needed to show equal expression. Figure 1b, the labeling of Jak1, Jak3 and Stat5 proteins in the Western blot panel would be presented more clearly if arrows were included to the designate bands. This will help the reader identify the bands, given that Jak1 expression is almost not visible. This will also benefit readers who have little experience with Jak/Stat Western blot patterns. Figure 1e, statistical analysis between the relative luciferase activity of WT/+IFNγ and ∆JH1/+IFNγ would add power to the conclusions. Figure 1e would be more aesthetically pleasing if the table for the panel on right looked similar to table on left. Figure 1e, left Western blot image, is ∆JH1/2 protein detectable within the WT/WT lane? Do the authors think the HA antibody is cross-reacting within this region? Figure Legend 1a, The readership would benefit from knowing the concentrations used to stimulate with IFNγ and IFNα in order to relate the findings in the context of the subsequent IFN stimulations. Figure Legend 1b and 1c. The readership would benefit from knowing the concentrations used to stimulate with IL-2 in order to relate the findings in the context of the subsequent dose curves performed in Figure 3b and Figure 5b as well as the following IL-2 stimulations. Figure Legend 1e, please indicate if the WBs shown are input obtained from unstimulated (basal) or stimulated with cytokine cell lysate. Figure 2a. Please indicate which residues are patient-derived Jak1 mutations by stating if they are shaded or unshaded. Figure Legend 2b. Please provide an explanation as to meaning of the different colors for WT (blue, orange, grey, yellow) and L633K (purple, green, dark blue and brown). If there are no differences, it would benefit the readers to use 1 color for WT and 1 color for L633K as the figure axis and titles are sufficient. Figure 2c, the authors should consider the use of statistical analysis between mutant V658F/WT and V658F+L633K/WT from non-stimulated cells which would help support conclusions on cis inhibition. Figure 2c, from the luciferase activity shown, in comparison to controls Jak3 R657Q mutation may not represent a true hyperactivating mutation. Are the authors concerned the readers may have a more conservative interpretation of Jak3 double mutant cis inhibition? Figure Legend 2c. The readership would benefit from knowing the concentrations used to stimulate with IL-2 in order to relate the findings to the global context of previous and subsequent IL-2 stimulations. Figure 2c. Given the large error bar associated with the double mutant, V658F L633K, it may be difficult for the reader to come to the same conclusion that the double mutant responds to cytokine stimulation similarly to WT Jak transfections. Figure 3b Panel 1 shows Stat5 luciferase activity according to the figure legend, and it is unclear if WB data immediately below are mislabeled as Stat1 instead of Stat5. If the labeling is accurate, why do the authors look at Stat1 instead of Stat5 in this experiment? Figure 3b, Please indicate if WBs represent samples from basal or plus IL2 and indicate the given concentration to help readers interpret Stat activation. Figure 3b, right Panel, the mutants are mislabeled between Jak1 and Jak3. Please flip labeling. Figure 3b. The labeling of Jak1, Jak3 and Stat1 should align with the protein band being designated. Alternatively, the authors can use arrows to point to the bands so that readers not familiar with Jak/Stat migration patterns are able to identify the proper band. Figure 3b. The reviewers are concerned with the lack of total Stat1 protein in the representative blot, and the impact this may have on luciferase assay. Figure 3b. Statistical analysis of basal versus IL2 treatments will help support that Jak1 I597F mutant responds to IL-2. Figure Legend 3b. Right panel, the readership would benefit from knowing the concentrations used to stimulate with IL-2 in order to relate the findings to the global context of previous and subsequent IL-2 stimulations. Figure 3c, readers unfamiliar with kinase assays would benefit from defining Apo control within the text. Figure 3c, Y-axis is lacking a label. Should it be labeled Tm (°C)? Figure 4a, The findings for trans inhibition over mutants would be strengthened by statistical analysis and benefit the reader in their interpretations. Figure 4a, The experiment would be strengthened by including a control lane with L633K/WT to see the effects of trans inhibition on WT IL2 induction. Figure 4a, Are the authors concerned that Jak1 WT/Jak3 WT shows poor induction in response to IL-2 within this system? Figure 4a, within the figure legend, please indicate whether the WB shown for pStat5 is representative of basal or IL2 stimulated cells. Figure 4a, statistical analysis would help the readers interpret the effects of L633K and homologous mutations on trans inhibition of overactive mutants. Also, are the authors concerned that readers interpretation may be impacted by low levels of Jak1 expression? Figure 4b, Please be consistent on labeling g or g throughout the figures and texts. Figure 4b and 4c are missing important reblots for Stat1. Interpretation of these data would be strengthened by the inclusion of a Stat1 total protein reblot. Figure 5a, left panel, this experiment should be repeated to increase the likelihood of detecting a significant change with a p value <.05 opposed to p=.07 value, which indicates a non-significant change. Figure 5a, left panel, labeling of Jak1, Jak3 and Stat5 should line up with the appropriately designated band. Again, alternatively, authors can use arrows to properly designate which band corresponds to the appropriate label. Are the authors concerned with the faint detection of Jak1 by WB and how this will impact readers interpretation of the data? Figure 5b, the authors claim that the K622A Jak1 mutant weakly responds to IL2 induction. Changing the Y axis to fold change may lend to easier interpretation of the data.This may actually show that the K622A Jak1 ATP mutant responds to IL2. Figure 5b, please indicate the concentration of IL2 in samples used for the WB. Figure 5b, Are the authors concerned with the reduced expression of Jak3 across transfection systems which may impact the luciferase activity readings?
Line 21: heteromeric partner JAK2/TYK2 JAK2 or TYK2 were both indispensable
Line 71: JH1-JH2 interphase interface but also
Line 83: The readership would benefit from a thorough list of JAK1 associated cytokines.
Line 85: Our results show that JAK1 have has varying
Line 209: Mutations in the JAK1 aαC-region including the
Line 230: pocket and have been veritably shown to inhibit ATP binding into JAK2 JH2 [36],.
Line 232: JAK1 K622A was chosen as its homologs has been shown to
Line 248 and 252: MgCl2 should be MgCl2
Line 267: U4Cgbγβ cells transfected with
Line 268: Below: pSTAT1, and HA-labeled cell lysates.
Line 278: and its JH2 in IL-2 and I IFNγ pathways,
Line 296: activation state in the FINAαR complex
Line 301: HA, and pSTAT5 antibodies were used
Line 332: dimerization interaction between JH2s domains may result in complete loss
Line 362: show that in the tetrameric IFNGγR complex
Line 368: whereas JAK1 phosphorylates the IFNGγR1
Round 2
Reviewer 2 Report
There are several typographical errors due to addition of the new text. These need to be addressed before publication.
Author Response
We thank the Reviewer for the comments, and have went through the text. Unfortunately, most of grammatical issues were not found in the Word-document where the track changed were shown, but in the PDF version. We apologize for the inconvenience of going through the apparently incorrectly transformed PDF version that included partial modifications instead of the final text. Our attempt was to download a clean PDF version and again apologize for the situation. However, we have now made the remaining corrections to the original Word-document and present both the Word document as well as PDF document with these changes.
Reviewer 3 Report
1) Many new errors were encountered within the text (too many to finish). This could possibly be due to our lack of understanding as to the presentation of the corrections. One example is line 95 where families is spelled "familiesy". A second example is line 102 where "These pathways" is written "these pathwasey". I have highlighted many of these instances within the first few pages but ultimately there were too many to continue. The author should take care to revise the manuscript for errors before submitting for final publication.
2) Figure 1a, Western blots for V658F Jak1/ WT Jak2 (STAT1 activity in gamma2A cells) is very different from the Western blot originally presented in manuscript.
3) Figure 1d, Jak1 image partially obscures the letter "d"
4) Figure 3b (left panel), labeling of IL-2 is not aligned beneath the image.
5) Figure 3b (right panel), Right image needs labeling of the y-axis and underneath the Western blot for IL-2 lanes.
6)Figure 5a, there is no need to show p=0.15
7)Figure 5b, Right panel is missing Jak labels.
Please see attachment for highlighted text and figures.
Of note, the condition Jak1 (WT) and Jak3 (I597F) were performed in both Figure 3b Left and Right panels, however, the Western blot bands for pStat5 look very different between the two. The authors draw a conclusion based on the image from the Left panel but do not take in to account that the same conditions used in the right panel does not support their conclusions regarding Jak1 I597F and basal pStat5 and Stat5 response to IL-2 (lines 278-287).

Author Response
We thank the Reviewer for the thorough revisions.
Please see the attachment for the answers.
